# Cardiovascular Toxicity Induced by Vascular Endothelial Growth Factor Inhibitors

**DOI:** 10.3390/life13020366

**Published:** 2023-01-29

**Authors:** Diana Mihalcea, Hayat Memis, Sorina Mihaila, Dragos Vinereanu

**Affiliations:** 1Cardiology and Cardiovascular Surgery Department, University of Medicine and Pharmacy Carol Davila, Splaiul Independentei 169, 050098 Bucharest, Romania; 2Cardiology Department, University and Emergency Hospital, 050098 Bucharest, Romania

**Keywords:** cancer, VEGF inhibitors, cardiovascular toxicity

## Abstract

Cardiotoxicity is an important side effect of vascular endothelial growth factor (VEGF) inhibitors therapy used in the treatment of various malignancies, leading to increased morbidity and mortality. Arterial hypertension, cardiac ischemia with the acceleration of atherosclerosis, arrhythmias, myocardial dysfunction and thromboembolic disease are the most feared cardiovascular adverse reactions due to VEGF inhibitors. Susceptibility for the occurrence of VEGF inhibitors-induced cardiotoxicity has multifactorial determinants, with a significant inter-individual variation. Baseline cardiovascular risk assessment of the patient, type and stage of cancer, dose and duration of VEGF inhibitors treatment and adjuvant chemotherapy or radiotherapy are the main predictors for cardiotoxicity. The role of the cardio-oncology team becomes essential for achieving maximum therapeutic anti-angiogenic effects with minimum cardiovascular side effects. This review will summarize the incidence, risk factors, mechanisms, management and treatment of VEGF inhibitors-induced cardiovascular toxicity.

## 1. Introduction

Cardiovascular disease and cancer are the two main causes of death worldwide [1]. Using advanced surgical techniques, chemotherapy and radiotherapy, the outcome of oncological patients has improved, making the disease curable [2]. Although highly efficient, chemotherapy has various side effects that alter quality of life [3,4]. Cardiotoxicity is the most feared adverse reaction, leading to premature morbidity and mortality in cancer survivors [4]. 

Tyrosine kinases, enzymes that phosphorylate tyrosine residue, are important mediators in various cell functions such as development, differentiation, metabolism and apoptosis. Due to overexpression, dysregulation and abnormal autocrine or paracrine stimulation, they may develop mutations and aberrant functions, leading to malignancy [5]. Tyrosine kinase inhibitors (TKIs), by reducing tyrosine phosphorylation due to competitive adenosine triphosphate (ATP) blocking, are potent agents used as targeted cancer therapy with a significant increase in survival [5]. These small molecules inhibit vascular endothelial growth factor (VEGF), epidermal growth factor (EGF), platelet-derived growth factor (PDGF) or fibroblast growth factor activity (FGF) [6,7]. TKIs can lead to cardiotoxicity due to various mechanisms such as endothelial lesion and atherosclerosis, metabolic impairment with decreased adiponectin and high glucose, altered nitric oxide (NO) synthase activity, glomerular dysfunction, and mast cell disruption [6,7].

In healthy subjects, VEGF plays an essential role in angiogenesis and normal development of the vascular system by several pleiotropic actions such as stimulating endothelial cell activity; forming, proliferating and migrating endothelial tubes; disincorporating the extracellular matrix; and increasing vascular permeability [7]. In the heart, VEGF was identified in the myofibroblasts from the myocardial infarction scar [7]. These cells have a significant role in the development, repair and positive remodeling of the cardiac tissue [6,7,8]. In addition, VEGF may aggravate the atherosclerotic process. In monocytes and macrophages, VEGF production is favored by incorporated oxidized low-density lipoprotein (LDL) and VEGF increases vascular permeability, including for LDL particles [6,7,8]. Furthermore, VEGF is also produced by macrophages, platelets, renal mesangial cells, osteoblasts, smooth muscle cells and keratinocytes and thereby has an important role in hematopoiesis, bone formations or injury healing [6,7,8]. 

In cancer patients, VEGF stimulates dysregulated neoangiogenesis, protects tumor cells from apoptosis, and increases the resistance to conventional oncological treatments; thus, VEGF has a pivotal role in tumor growth, proliferation and spread [7,8]. Additionally, conventional chemotherapy and radiotherapy increase VEGF production in the tumor cells [6]. Thereby, inhibition of VEGF or its receptors (VEGFRs) targets not only the pro-angiogenic effect of VEGF, inhibiting neovascularization, but also the pro-survival and anti-apoptotic action of VEGF [6,7,8,9]. Moreover, the association between VEGF inhibitors and standard cytotoxic drugs or radiotherapy may lead to a stronger antitumor effect since a VEGF blockade makes neoplastic cells more susceptible to treatment [6,9]. 

VEGF is a family with seven members; isoform VEGF-A is the most studied, having a central role in angiogenesis [7]. Four spliced forms for VEGF-A isoform are known (VEGF_121_, VEGF_165_, VEGF_189_ and VEGF_206_), but VEGF_165_ is the most widespread in a variety of human solid tumors [7]. After being secreted from the tumor, VEGF binds to its receptor (VEGFR-1 or VEGFR-2) from endothelial and bone-marrow derived cells [6,7,8]. While the role of VEGFR-1 is not fully established, VEGFR-2 mediates the angiogenic effects of VEGF-A isoform [7,8,9]. Thus, VEGF-A and VEGFR-2 are the main targets for antiangiogenic therapies [6,7,8,9]. 

In clinical practice, four types of VEGF inhibitors are used: monoclonal VEGF antibodies (bevacizumab), monoclonal VEGFR antibodies (ramucirumab), soluble “trap” receptors (aflibercept) and VEGFR-TKIs (sunitinib, sorafenib, pazopanib, vandetanib, axinitib, regorafenib) [8]. These drugs are used successfully in different malignancies: glioblastoma, metastatic renal, colorectal, thyroid, non-small cell lung and gastric cancer, sarcoma, pancreatic neuroendocrine tumors, hepatocellular carcinoma and genital cancer [4,10]. 

Theoretically, because angiogenesis is the central process for tumor progression and has a limited role in healthy cells, the probability of VEGF inhibitors for side effects is low [7]. However, the specificity of VEGF inhibitors is not absolute; they may have off-target effects on cardiovascular tyrosine kinases and thus favor the occurrence of cardiotoxicity [11]. Moreover, knowing the essential role of VEGF in regulating endothelium-dependent vascular tone and the influence of vasculature for heart function, cardiovascular side effects of anti-angiogenesis agents can be explained [6,10]. On the contrary, resistance to VEGF and VEGFR-TKIs is significantly lower than in traditional chemotherapy [7]. 

This review will focus on the incidence, risk factors, mechanisms, management and treatment of VEGF inhibitors-induced cardiotoxicity. 

## 2. VEGF Inhibitors-Induced Cardiovascular Toxicity

Cardiovascular side effects of VEGF inhibitors can be classified as follws: arterial hypertension; cardiac dysfunction and heart failure; cardiac ischemia; arrhythmias and QT interval prolongation; thromboembolic disease.

### 2.1. Definition and Classification

Identified since the 1960s as “toxicity that affects the heart”, cardiotoxicity is a serious side effect of chemotherapy, with forms ranging from asymptomatic to severe cardiac dysfunction and irreversible heart failure [3,4,12]. Up to 30% of cancer survivors may develop a cardiac adverse reaction secondary to cytotoxic drugs [3]. Echocardiography is the most used method for the assessment of patients before, during and after chemotherapy, and left ventricular ejection fraction (LVEF) represents the standard parameter for diagnosis of cardiotoxicity [13]. However, other echocardiographic parameters, such as tissue velocity imaging and myocardial deformation, are more accurate for the early detection of cardiac dysfunction, before the alteration of LVEF [2,3,4]. Global longitudinal strain (GLS) is also an independent marker that can predict the subsequent decrease of LVEF [10]. A reduction of more than 15% of GLS from the baseline value allows an early detection of subclinical cardiotoxicity [3].

Furthermore, cardiac biomarkers, such as troponin I, a marker of myocardial injury and N-terminal pro-brain natriuretic peptide (NT-pro-BNP), a marker of elevated filling pressure, may be adjuvant tools for an early diagnosis of chemotherapy-induced myocardial dysfunction [3,4]. 

In 2022, new cardio-oncology guidelines divided cardiac dysfunction related to chemotherapy into symptomatic and asymptomatic forms [4]. Symptomatic cardiotoxicity may vary from mild to moderate, severe and very severe depending on the gravity of heart failure symptoms, the need for intravenous therapy and hospitalization [4]. Asymptomatic cardiotoxicity is classified as severe, moderate and mild [4]. If severe asymptomatic myocardial dysfunction requires a decrease of LVEF below 40%, the moderate form is defined by a LVEF reduction between 40–49% associated with a decline in GLS by more than 15% or a rise of cardiac biomarkers [4]. In addition, the mild form of asymptomatic cardiotoxicity involves a subclinical dysfunction, with GLS decreasing by more than 15% or a rise of biomarkers despite a preserved LVEF [4]. 

Depending on timing, cardiac dysfunction can be classified as acute, subacute or chronic. Acute cardiotoxicity occurs in the first two weeks after chemotherapy started, subacute cardiotoxicity after two weeks and chronic cardiotoxicity in the first year (early chronic cardiotoxicity) or after one year (late chronic cardiotoxicity) [2]. Often, chronic cardiotoxicity leads to systolic myocardial dysfunction with irreversible heart failure and death [2,3,4]. 

The diagnosis of immune inhibitors induced-myocarditis is established by clinical criteria and, in selective cases, by pathohistological findings using endomyocardial biopsy [4]. Clinical diagnosis requires elevation of troponin associated with one major criterion or two minor criteria, after infectious myocarditis and acute coronary syndrome are excluded [4]. Major criteria are assessed by cardiac magnetic resonance (CMR) with Lake Louise T1 and T2 pathological changes, increased extracellular volume or late gadolinium enhancement [4]. Minor criteria include clinical symptoms, ventricular arrhythmias, cardiac arrest or conduction system disorders, reduction of left ventricular (LV) systolic function with diffused or regional wall motion, incomplete Lake Louise CMR criteria and associated immune disorders, such as myasthenia gravis, myopathy or myositis [4]. 

### 2.2. Risk Factors

The occurrence and severity of cardiotoxicity have an inter-individual variation, resulting from the interaction between genetic and phenotypic risk factors [2,3,4]. The main clinical and demographic risk factors that predispose to cardiotoxicity are presented in Table 1. 

### 2.3. Arterial Hypertension 

Arterial hypertension is the most common cardiovascular side effect of VEGF inhibitors, but also it can be an important comorbidity in cancer patients [4,14]. Hypertension incidence reaches up to 45%, with severe forms in 20% of patients [3,4]. When VEGF-TKIs are combined, the risk for hypertension exceeds 90% [8]. The onset of hypertension is variable, between the first hours after initiation of therapy and 1 year [4,5]. The severity of arterial hypertension is higher in patients with baseline cardiovascular risk factors, renal cell carcinoma, high doses of VEGF inhibitors and concomitant administration of other cancer drugs (Table 1) [4,15]. 

The cellular mechanisms involved in the occurrence of hypertension induced by VEGF inhibitors are incompletely elucidated. VEGF stimulates NO release, with a vasodilator effect, by activating NO-synthase activity [5]. In patients with VEGF inhibitors, the serum NO level is reduced but reversible after treatment discontinuation [6]. In addition, VEGF-TKIs determine an increased excretion of endothelin-1, with a powerful vasoconstrictor effect [6]. In animal models, using an endothelin-1 receptor antagonist, the acute hypertension induced by VEGF inhibitors was reversed successfully [5,15]. More than that, during VEGF inhibitors therapy, rarefaction, a reduction of capillary density, was identified [6]. So far, it is not known if rarefaction is a cause or a consequence of hypertension [6]. Other possible mechanisms of hypertension include high oxidative stress and inhibition of Ca^2+^ channel activation, which favor vascular constriction [8]. In addition, in kidneys, VEGF inhibitors cause glomerular endotheliosis and thrombotic microangiopathy, leading to proteinuria, increased serum creatinine and hypertension; a high sodium diet may exacerbate their severity [8,16]. The role of the renin-angiotensin-aldosterone system in arterial hypertension induced by VEGF inhibitors is minor compared with essential hypertension [7]. In experimental models, VEGF-TKIs correlate with a decrease in renin activity, but angiotensin-converting enzyme (ACE) inhibitors have a limited impact on hypertension compared with Ca^2+^-channel blockers [6]. Consistent with these findings, in patients with hypertension secondary to sunitinib, serum renin decreased by 60%, but aldosterone remained unchanged [6,7]. 

Before initiation of VEGF-TKIs, assessment and correction of cardiovascular risk factors, including preexisting hypertension, are mandatory [4]. Identifying concomitant medication (non-steroidal inflammatory drugs, steroids, erythropoietin) that can cause or aggravate hypertension is required [3]. During VEGF inhibitors therapy, blood pressure (BP) should be measured frequently, at each cardio-oncological visit [4]. Home BP monitoring is also recommended, but is not always feasible [4,5]. An acute rise of BP induced by VEGF inhibitors can precipitate an acute target-organ complication, such as stroke, myocardial ischemia, pulmonary edema or acute kidney injury [6]. Maintaining BP at optimum values, below 130/80 mmHg in high cardiovascular risk patients and otherwise below 140/90 mmHg prevents the occurrence of chronic renal, cerebrovascular and cardiac dysfunction, and thus decreases morbidity and mortality [4,17].

Patients with BP consistently exceeding 140/90 mmHg require antihypertensive treatment. The choice of optimal therapy follows international guidelines for arterial hypertension [16]. ACE inhibitors, angiotensin receptor blockers (ARBs) and non-dihydropiridine Ca^2+^-channel blockers are first-line treatments [4,16,18]. In patients with heart failure or cardiac dysfunction, ACE inhibitors and beta-blockers are the recommended antihypertensive drugs [4,19]. Of beta blockers, nebivolol is an important option due to its additional effect of increasing NO release, blocked by VEGF inhibitors therapy [3]. As NO donors, nitroglycerin and sodium nitroprusside are preferred intravenous drugs for hypertensive emergencies [19]. Diltiazem and verapamil should be avoided due to interaction with VEGF inhibitors in cytochrome P4503A4, resulting in increased plasma levels of antineoplastic drugs [3,6]. Because VEGF inhibitors may determine diarrhea and dehydration, diuretics should be carefully used, considering also their risk of electrolyte disturbance and QT interval prolongation [3,4]. 

Usually, the occurrence of hypertension does not require VEGF inhibitors withdrawal; moreover, antihypertensive drugs do not alter the potency of VEGF-TKIs [4,10]. However, in severe or uncontrolled hypertensive patients, reduction or discontinuation of chemotherapy may be considered [4]. Once BP is optimal, VEGF inhibitors can be restarted in order to obtain the maximum anticancer effect [4,14]. 

Presented in less than 1% of patients, a severe side effect of VEGF inhibitors and associated hypertension is *posterior reversible leukoencephalopathy*, manifested by nausea, headaches, confusion, epileptic seizures, visual disorders, focal neurological deficit and, finally, a coma [6,20].

Brain magnetic resonance imaging identifies disseminate bilateral white matter abnormalities, typically with changes in the posterior fossa in T2 sequences [20,21]. Cellular mechanisms involve alteration of cerebral endothelial permeability and impaired vascular autoregulation, with important edema [20]. Early diagnosis, urgent treatment of hypertension and discontinuation of VEGF-TKIs lead to the reversibility of the disease and a favorable prognosis; otherwise, cerebral ischemia or bleeding may occur [6,20,21]. 

It is important that every patient benefits from individualized treatment from the cardio-oncological team, taking into account the type and severity of cancer, alternative efficacy therapy and coexistence of cardiovascular risk factors [4,6,9]. 

### 2.4. Cardiac Dysfunction and Heart Failure

VEGF inhibitors-induced myocardial toxicity ranges from asymptomatic forms to irreversible heart failure, cardiogenic shock and death [4,6]. The incidence of cardiac dysfunction varies between 3 and 15% and heart failure may occur in 10% of the patients [3,4,8]. However, increased NT-pro-BNP (over 300 pg/mL) was identified in 21% of patients, but only 9% associated a decrease of LVEF with more than 10% [22]. Myocardial dysfunction can develop early, in the first days after therapy initiation or several months later [3]. The risk for cardiotoxicity is higher in patients with preexisting hypertension, diabetes, coronary artery disease, atherosclerosis and previous chemotherapy or radiotherapy (Table 1) [3,4,8]. No relationship between the total dose and duration of chemotherapy and cardiac dysfunction has been reported yet [6]. 

Cardiac dysfunction may be secondary to vascular toxicity due to VEGF inhibitors, by increased arterial stiffness and altered ventricular-arterial coupling with LV hypertrophy and myocardial ischemia [22,23]. However, a direct cytotoxic effect of chemotherapy cannot be excluded [6]. VEGF has an essential role in cardiac response to arterial hypertension and VEGF-TKIs accelerate the decompensation of hypertensive heart disease to dilatation and heart failure [3,4,8]. Moreover, the highest incidence of cardiotoxicity is reported in thyroid cancer patients with iatrogenic hypothyroidism, induced by VEGF inhibitors [9]. A low level of T3 hormone induces endothelial dysfunction with vasoconstriction, and reduces cardiac output with an unfavorable prognosis [9]. There is no direct evidence that hypothyroidism is a cause or an aggravating factor for myocardial dysfunction secondary to VEGF-TKIs [9].

The cellular mechanism of VEGF-TKIs-induced cardiotoxicity involves mitochondrial dysfunction and inhibition of the AMP-kinase pathway; no apoptosis or fibrosis was found by endomyocardial biopsy, suggesting that cardiac dysfunction due to VEGF inhibitors is a reversible process [5,23,24]. Nevertheless, in animal models, sunitinib favors cardiomyocyte apoptosis by increased caspase activity and additionally inhibits platelet-derived growth factor receptors, leading to decreased myocardial pericytes, microvascular density and contractile function [9,25]. 

Due to metastatic cancer and low life expectancy, no long-term follow-up for these patients is reported. However, optimal treatment of arterial hypertension decreases the risk of cardiac dysfunction and heart failure [16,26]. In heart failure patients, an intensive and appropriate therapy with ACE inhibitors or angiotensin receptor/neprilysin inhibitors, beta-blockers, diuretics and sodium-glucose cotransporter-2 inhibitors may slow the progression of the disease [4,26].

For prevention or early diagnosis of cardiac dysfunction in low-risk patients, echocardiography and an electrocardiogram (ECG) are recommended only at baseline, while in moderate-risk patients cardiology evaluation has to be carried out every four months [4]. Patients with a high and very high risk of cardiac dysfunction require a cardiovascular evaluation every three months [4]. In patients with significant risk factors receiving sunitinib, sorafenib or pazopanib, early evaluation after two weeks may be considered [6]. 

The occurrence of myocardial dysfunction induced by VEGF-TKIs force the interruption of chemotherapy and the introduction of protective cardiovascular therapy with beta-blockers and ACE inhibitors [3,4]. If cardiotoxicity is reversible, the resumption of VEGF inhibitors may be considered [4,6]. In patients with significant LV dysfunction at baseline, alternative chemotherapy to VEGF inhibitors should be recommended [6]. Thus, the role of the cardio-oncological team is essential for monitoring, prevention and early diagnosis of cardiac dysfunction induced by VEGF-TKIs. 

### 2.5. Cardiac Ischemia

Myocardial ischemia secondary to VEGF-TKIs, ranging from asymptomatic forms to acute coronary syndromes, has a low incidence: 1.4% for sunitinib, 1.7% for sorafenib and less than 1% for bevacizumab in breast cancer patients, but increased to 3.8% in metastatic disease [4,5]. The main risk factors for the development of cardiac ischemia are as follows: preexisting hypertension, atherosclerosis, diabetes, increasing age, lower body mass index, history of cardiac disease or heart failure, association with other chemotherapeutic agents or radiotherapy (Table 1) [3,4]. Moreover, coronary artery disease and arterial hypertension are the most important predictors of the occurrence of heart failure in cancer patients treated with VEGF inhibitors [9,27].

In addition to the procoagulant effect induced by the neoplastic disease itself, there are several mechanisms by which VEGF-TKIs cause myocardial ischemia: reduced nitric oxide synthesis, endothelial injury, arterial inflammation and premature atherosclerosis, vasoconstriction and platelet reactivity [4,8,9]. In animal models, VEGF inhibitors induce accelerate atherosclerosis, but without increased plaque vulnerability, suggesting that antiangiogenic drugs favor plaque stabilization [28]. However, when autocrine the VEGF signaling pathway is insufficiently inhibited in endothelial cells, plaque erosions, endothelial injury and acute arterial thrombosis are found [8,29]. Sunitinib determines an abnormal vasoreactivity of coronary microvasculature, with a significant reduction of coronary flow reserve and myocardial function [27]. Moreover, inhibition of PDGF leads to pericytes depletion, which causes significant endothelial dysfunction with impairment of coronary microcirculation and, finally, myocardial damage [9,27]. These two mechanisms, called the pericyte-endothelial-myocardial coupling, are responsible for the “vascular” cardiotoxicity of VEGF inhibitors [9,27,29].

The diagnosis of previous cardiac ischemia is essential before initiating VEGF-TKIs [3,4]. Moreover, preexisting coronary artery disease significantly increases the risk of VEGF-induced myocardial ischemia [4]. The diagnosis and treatment of chronic and acute coronary syndromes after VEGF inhibitors therapy follow the current guidelines [30,31,32,33]. In addition, stress echocardiography may be used to diagnose the impairment of coronary microcirculation and perfusion due to bevacizumab, sorafenib or sunitinib in patients with intermediate or high-pretest probability of coronary artery disease [14]. 

In mice with myocardial infarction, sorafenib decreases the cardiac repair process by c-kit+ stem cell apoptosis [9]. Adding beta-blockers to sorafenib-treated mice reduces myocytolysis with improved outcomes and decreased mortality [9,34]. After an acute cardiovascular event, current guidelines recommend discontinuing any stressor, including chemotherapeutic agents, for at least sixty days [7,35]. Statins, besides the essential cardiovascular effects, reduce VEGF synthesis in microvascular endothelium and upregulate VEGF bioavailability in macrovascular endothelial cells [18]. Moreover, low doses of statins are proangiogenic, whereas high doses become anti-angiogenic [18]. Thus, statins should be used in high concentrations in cancer patients with cardiovascular risk factors, receiving chemotherapy with VEGF inhibitors [18]. 

In addition, the fact that VEGF-TKIs are associated with a high risk of thrombocytopenia and bleeding represents a challenge for the medical and interventional treatment of myocardial ischemia [4,31,32,33]. The use of antiplatelet drugs and anticoagulants should be made with caution with a minimal duration of dual antiplatelet therapy [3,33]. In highly symptomatic patients, depending on frailty, the least invasive procedure is recommended [4,33]. 

However, management and treatment decisions of VEGF inhibitors-induced cardiac ischemia should be individualized according to cancer severity, life expectancy, comorbidities, or alternative highly efficient chemotherapeutic agents [4,14,32]. 

### 2.6. Arrhythmias and QT Interval Prolongation

Brady- or tachyarrhythmias, and conduction defects may be present before starting chemotherapy in 16–36% of cancer patients. [3,4]. During VEGF inhibitors therapy, arrhythmias can be related mainly to QT interval prolongation, but cardiac ischemia, myocardial dysfunction and predisposing factors may be incriminated [3,4,5]. 

QT interval prolongation can be caused by VEGF-TKIs, concomitant medication, extracardiac diseases, or electrolyte disturbances (Table 2) [3,4]. The incidence of QT interval prolongation varies with individual VEGF inhibitors, with a maximum of 8% for vandetanib [4,5,6]. QT prolongation favors the occurrence of life-threatening arrhythmias, such as torsade de pointes [4,5,6]. The mechanism for QT interval prolongation involves the activation of cardiomyocyte potassium channel proteins [6,36]. 

The QT interval should be assessed before and during VEGF inhibitors therapy [3,5]. A 12-lead electrocardiogram is recommended at baseline, 1–2 weeks after initiation or dose modification, monthly in the first 3 months and then periodically depending on the risk factors of the patient and chemotherapeutic regimen [3,4,5]. For vandetanib, the QT interval will be monitored at baseline, 2–4 weeks, and 8–12 weeks after starting treatment and then every 3 months [37]. A QT interval of 450 ms in men and 460 ms in women is considered the upper limit of the normal value [3]. A QT prolongation of more than 500 ms or an increase from baseline of more than 60 ms represents an important risk factor for torsade de pointes [4,5]. This significant finding requires temporary cessation of antiangiogenic drugs and prompt correction of the QT interval prolongation-associated risk factors (Table 2) [3,38]. After normalizing the QT interval, an effective chemotherapeutic alternative to VEGF-TKIs should be identified [4]. If not, VEGF inhibitors therapy may be resumed, but with a reduced dose and increased frequency of electrocardiograms for monitoring the QT interval [3,4]. 

The occurrence of torsade de pointes is scarce [3,4]. Usually, its treatment consists of intravenous magnesium sulphate; if arrhythmia persists, overdrive transvenous pacing or isoprenaline are required [3,39]. Urgent defibrillation must be performed in sustained ventricular arrhythmias with hemodynamic instability [3,4,39].

Ponatinib, sorafenib, and sunitinib increase the risk of the development of atrial fibrillation [3]. The management of this supraventricular arrhythmia should follow current guidelines, with rhythm or rate control and thromboembolic prophylaxis [3,40,41]. However, anticoagulation for atrial fibrillation in active malignancy is challenging [3,4]. On the one hand, cancer patients have a prothrombotic state and a high bleeding risk, and on the other hand, CHA_2_DS_2_-VASc and HAS-BLED risk scores are not validated in this population [3,42]. However, anticoagulation may be considered when the platelet count exceeds 50,000/mm^3^ and the CHA_2_DS_2_-VASc score is at least 2 [3]. Anticoagulation options are therapeutic low molecular weight heparin (LMWH), vitamin K antagonist (VKA) when the international normalized ratio is a stable or non-VKA oral anticoagulant (NOAC) [3,40,41]. In metastatic disease with high bleeding risk, the preferred option is LMWH and warfarin should be avoided [3]. Even if recent studies suggest that NOAC is safe in cancer patients, with lower bleeding and thromboembolic risk compared with VKA, the number of patients with active malignancy was small or cancer was an exclusion criterion in these trials [41,42,43]. 

Overall, monitoring the QT interval, correction of associated QT prolongation risk factors and antiarrhythmic and antithrombotic therapy in cancer patients with VEGF inhibitors-induced arrhythmias requires a dedicated cardio-oncology approach with personalized management for each patient [3]. 

### 2.7. Thromboembolic Disease

Active malignancies trigger coagulation through several pro-inflammatory and pro-angiogenic cytokines, resulting in a procoagulant, pro-aggregating and antifibrinolytic environment [3,4]. 

#### 2.7.1. Arterial Thrombosis 

Arterial thrombosis appears in only 1% of cancer patients [3,19]. However, in metastatic pancreatic, breast, lung or colorectal neoplasms treated with VEGF inhibitors, anthracyclines or taxanes, the incidence of arterial thrombosis increases significantly and it is associated with a poor prognosis [3,17]. The risk factors, mechanisms and management for cerebrovascular events are the same as for myocardial ischemia due to VEGF inhibitors, presented above. Compared to cardiac thrombosis, cerebrovascular disease secondary to VEGF-TKIs is rare, with a mean incidence of 0.5% for ischemic stroke and 0.3% for hemorrhagic events [17,19]. The highest incidence was found for cediranib (3.2%) and pazopanib (2.4%) [18,44]. 

#### 2.7.2. Venous Thrombosis and Thromboembolism (VTE) 

Venous thrombosis and thromboembolism (VTE) are more common than arterial thrombosis in cancer and can reach up to 20% of hospitalized patients [3]. VTE is favored by active malignancy, type of chemotherapy, drug administration (venous catheters) or patient’s risk factor for thrombosis [3]. The risk for venous thrombosis and recurrent episodes induced by VEGF inhibitors is increased six-fold and two-fold, respectively, with a maximum incidence of 11% [3,45,46]. Incidental VTE is common in cancer patients, but it requires similar management to symptomatic episodes, due to the high risk of recurrence and mortality [3,46]. 

Treatment of VTE due to malignancy or chemotherapy includes LMWH for the first six months, with the possibility of extending the therapy until the cancer is cured or for an indefinite period [45,47]. In cancer patients, LMWH is superior to VKA, with no difference regarding bleeding events or mortality [3,45]. Among NOACs, edoxaban and rivaroxaban may be long-term alternatives to LMWH in patients without gastrointestinal neoplasia, renal impairment or intake and absorption difficulties [45]. In patients with an active hemorrhage or high bleeding risk, when anticoagulation is contraindicated, retractable or definitive venous filters could be an alternative [3,45]. Association between heparin and venous filters is not routinely recommended in cancer patients [3,45]. In active malignancies with hemodynamically unstable VTE, due to increased mortality, thrombolysis may be considered in particular cases [3]. However, brain tumors, metastatic cancer, life expectancy or high bleeding risk are significant contraindications for fibrinolytic treatment [3,45]. Surgical embolectomy is an alternative, but it is associated with increased morbidity [3,45]. 

## 3. Conclusions

Despite the increased efficacy for treating various malignancies, VEGF inhibitors associate with a significant risk of cardiovascular toxicity, leading to high morbidity and mortality in these patients. For primary prevention or early treatment of VEGF inhibitors cardiovascular side effects, a rigorous baseline cardiovascular risk assessment is mandatory. The baseline cardiovascular evaluation should consist of a clinical examination, including blood pressure measuring, an ECG, echocardiography, routine blood tests and cardiac biomarkers troponin and NT-pro-BNP. In patients with moderate and high cardiovascular risk, periodic cardiac evaluation is necessary during treatment, at least at three to six months, but also after oncological treatment is finished, for preventing late cardiovascular toxicity. The guideline recommendations should be followed, but always adapted to the patient, his cardiovascular risk and associated diseases. Whenever possible, optimal doses of VEGF-TKIs should be continued in association with cardiovascular drugs. An interdisciplinary approach by the cardio-oncology team allows the proper management of cancer patients to achieve the maximum therapeutic effect from anti-angiogenic agents while minimizing cardiovascular toxicity. 

## Figures and Tables

**Table 1 life-13-00366-t001:** Risk factors for chemotherapy-induced cardiotoxicity (modified from [3,4]).

Demographic Risk Factors	Age (Pediatric and Elderly Population)
Lifestyle risk factors	− Sedentary − Obesity − High alcohol intake
2. CV risk factors	− Smoking − Arterial hypertension − Diabetes mellitus − Dyslipidemia − Family history of premature CV disease (<50 years)
3. Myocardial disease	− Heart failure − Asymptomatic LV dysfunction (EF < 50%) − Coronary artery disease (angina, myocardial infarction, PCI, CABG) − Moderate or severe valvular heart disease − Cardiomyopathy (hypertrophic, dilated, restrictive) − Hypertensive heart disease with LV hypertrophy − Significant sustained arrhythmias
4. Previous cancer treatment	− Anthracyclines − Radiotherapy to chest or mediastinum

CABG = coronary artery bypass graft; CV = cardiovascular; EF = ejection fraction; LV = left ventricle; PCI = percutaneous coronary intervention.

**Table 2 life-13-00366-t002:** Risk factors for QT interval prolongation in cancer patients (modified from [3,4]).

*Correctable*	*Non-Correctable*
Electrolyte disturbances (vomiting, diarrhea, loop diuretics) − Hypokalaemia − Hypomagnesaemia − Hypocalcaemia	Family history of sudden death Personal history of syncope Baseline QT interval prolongation Female gender Advanced age Heart disease Impaired renal function Impaired hepatic drug metabolism
2. Extracardiac disease − Hypothyroidism	
3. Concomitant medication − Antiarrhythmic − Antibiotic − Antifungal − Antidepressant − Antiemetic − Antihistamine	

## Data Availability

Not applicable.

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
