# Peer review of "Cardiovascular Toxicity Induced by Vascular Endothelial Growth Factor Inhibitors"

_life, 2023, doi:10.3390/life13020366_

Round 1

Reviewer 1 Report

The authors reviewed the importance of the side effects of VEGF inhibitors as treatment for several diseases. They summarized the incidence, risk factors and some mechanisms of VEGF inhibitors-induced cardiovascular toxicity.

In fact, the manuscript is well-written but has some issues. Please correct some spelling errors and some language mistakes.

Author Response

We appreciate the opportunity to review our paper ‘"Cardiovascular toxicity induced by vascular endothelial growth factor inhibitors" and we are glad that you consider it as a possible publication in Life journal. 

First of all, we would like to thank you for your comments and suggestions, which have helped us in improving significantly our original submission. We corrected all spelling errors and the language mistakes. 

We decided to mark the changes in the revised manuscript in yellow, in order to make their identification easier.

We would like to apologize for the relative delay in submitting the revised manuscript, but we have tried to address carefully all the issues addressed.

Reviewer 2 Report

Diana Mihalcea wrote an interesting paper entitled „Cardiovascular toxicity induced by vascular endothelial growth factor inhibitors.” The authors carefully analyze the cardiovascular side effects encountered after the treatment with various VEGF inhibitors used in cancer therapy. They also describe how cardiovascular drugs, such as beta-blockers, statins, platelet inhibitors, NOAC, heparin, etc., help minimize the harmful effects of VEGF inhibitors. The paper would be of interest to the Life readers after the application of some major and minor comments.

Major comments

·        Add more information about the VEGR role and its release mechanism. You have to underline its impact on endothelial cells, which are essential for the cardiovascular system and angiogenesis. Explain why the inhibition of VEGF is so important to stop tumor growth. It is crucial to write about the physiological function of VEGF to understand better the cardiovascular side effect such as hypertension, cardiac ischemia, atherosclerosis, arrhythmias, myocardial dysfunction, and thromboembolism. Similarly, you must explain tyrosine kinase function and the connection between its blockade, tumor growth, and subsequent cardiovascular toxicity. You have to make an additional paragraph about this basic knowledge or put it in the introduction. This basic knowledge is scattered throughout the publication, but it is crucial to be good elucidated at the beginning of the manuscript and then only marginally mentioned. 

·        It will be a good summary of the authors' proposal on what clinical and laboratory parameters should be performed periodically in patients treated with VEGF inhibitors to diagnose cardiovascular problems early and start treating them as soon as possible.

Minor coments

Abstract

·        Explain VEGF

·        Add to your affiliation the department or clinic (cardiology, oncology, or others). Now there are only your universities.

Introduction

·        Add the abbreviations

epidermal growth factor (EGF)

fibroblast growth factor (FGF)

·        VEGF or its receptors (VEGFRs)

VEGF inhibitors-induced cardiovascular toxicity

Definition and classification

·        alteration of LVEF [2-4} = [2-4]

Author Response

We appreciate the opportunity to review our paper ‘"Cardiovascular toxicity induced by vascular endothelial growth factor inhibitors" and we are glad that you consider it as a possible publication in Life journal. 

First of all, we would like to thank you for your comments and suggestions, which have helped us in improving significantly our original submission. 

We decided to mark the changes in the revised manuscript in yellow, in order to make their identification easier.

We would like to apologize for the relative delay in submitting the revised manuscript, but we have tried to address carefully all the issues addressed.

New information was added in “Introduction” about the essential role of VEGF for cardiovascular system, with pleiotropic effects on endothelial cells and angiogenesis and positive cardiac remodeling after myocardial infarction. In addition, the role of VEGF in tumor growth and the consequences of VEGF inhibition in cancer were presented.

New paragraph with the role of tyrosine kinase in normal cells and malignancy was added, also with the main mechanisms of cardiotoxicity induced by tyrosine kinase inhibitors. Indeed, we focused in this review only on cardiovascular toxicity due to one class of tyrosine kinase inhibitors – VEGF inhibitors.

We also added in the “Conclusion” a summary of a cardiac consultation during cardio-oncology interdisciplinary evaluation, with respect of the current guidelines but also taking into account the particularities of each patient, with his cardiovascular risk factors and associated diseases.   

All minor comments were also updated, as suggested.